# A Systematic Review of the Implementation and Effectiveness of ‘The Daily Mile’ on Markers of Children’s Health

**DOI:** 10.3390/ijerph20136203

**Published:** 2023-06-22

**Authors:** Luke Hanna, Con Burns, Cian O’Neill, Edward Coughlan

**Affiliations:** Department of Sport, Leisure and Childhood Studies, Munster Technological University, Bishopstown, T12 P928 Cork, Ireland; con.burns@mtu.ie (C.B.); cian.oneill@mtu.ie (C.O.); edward.coughlan@mtu.ie (E.C.)

**Keywords:** health, physical activity, initiative, implementation process

## Abstract

Currently, a high percentage of children globally fail to meet the World Health Organisation’s (WHO) recommended daily physical activity (PA) guidelines. The Daily Mile (TDM) is a school-based PA initiative, designed to improve primary school children’s PA behaviour. The purpose of this review was to evaluate the extant TDM implementation process and identify its impact on health-related metrics. Three databases were used to search for articles from the time TDM originated in 2012 until February 2022. The identification and screening process of articles for their ability to meet this review’s eligibility criteria were facilitated by use of PRISMA and Rayyan. Sixteen articles from the initial search (n = 202) were deemed eligible for inclusion. An analysis of these articles identified five common outcome categories that permeated throughout the research articles: (1) cardiorespiratory fitness (CRF); (2) anthropometry and body composition; (3) PA; (4) cognition; and (5) process evaluation. Results presented from the included articles suggests TDM positively impacts markers of a variety of health-related metrics, namely CRF and PA. However, implementation barriers including TDM’s repetitive nature, time constraints associated with competing curriculum demands and inadequate facilities regularly necessitate the adaptation and development of the original TDM format by schools and teachers.

## 1. Introduction

Physical Activity (PA) has been shown to positively impact markers of children’s health and wellbeing [1,2]. Moreover, a physically active lifestyle originates in early childhood and has been reported to impact PA behaviour during the transition from adolescence to adulthood [3]. Conversely, physical inactivity has emerged as a global pandemic in recent times, whereby sedentariness has been demonstrated as having a detrimental impact on the health of children, adolescence and adults [4]. According to the World Health Organisation (WHO), over 340 million children and adolescents were either overweight or obese globally in 2016 [5]. Engaging in regular PA has been identified as an important factor associated with preventing [6] and treating childhood obesity [7].

The WHO have recommended that children aged between 5–17 years old should engage in PA of a moderate–vigorous intensity (MVPA) for at least 60 min every day, in addition with specifically incorporating aerobic activities of a vigorous intensity (VPA) and bone- and muscle-strengthening activities for a minimum of 3 days each week [7]. In support, the international analysis of PA guidelines conducted by Parrish et al. [8] revealed how most countries advocate children’s daily PA to consist of a moderate–vigorous intensity. The intensity of PA can be measured by using a ratio of an individual’s working metabolic rate in comparison to their resting metabolic rate known as metabolic equivalents (MET’s), whereby moderate PA (MPA) requires 3.0–6.0 METs and VPA requires > 6.0 METs [9].

In a study conducted by Gomes et al. [10], the MVPA behaviour of a sub-sample of children aged 9–11 years (n = 6553) from 12 countries including India, China, USA, South Africa, and the United Kingdom (UK) was examined. It was reported that only 4.8% of children achieved the minimum recommended 60 min of MVPA across each day of a 7-day week [10]. Moreover, a higher percentage of boys (7.6%) were reported to meet PA guidelines across 7 days of a week when compared to girls (2.4%) [10]. In Ireland, the Children’s Sport Participation and Physical Activity Study [11] determined that a mere 17% of primary school children were meeting the daily PA recommendations, whereby a higher percentage of primary school boys (23%) were reported to meet these guidelines when compared to primary school girls (13%). These findings highlight the urgent need to identify and support initiatives that can bridge the gap between PA and physical inactivity among children, thereby aspiring to reduce the health risks associated with same for the children of today’s, and future, generations.

Primary schools should be considered suitable settings to implement PA initiatives, as children in Ireland and the UK are reported to attend for approximately half the days of a typical calendar year [12,13]. According to Sturm et al. [14], the school system in the USA and UK recommend that children engage in at least 30 min of MVPA during every school day. Subsequently, the WHO [15] have advocated for an increase in school-based initiatives that serve to influence educational policy to tackle the concerning issues associated with physical inactivity in children and adolescents.

Spanish primary school children (n = 900, aged 10–12 years) engaged in higher levels of PA during school when compared to the PA they engaged in during sport and leisure-time activities [16]. Burns et al. [17] meta-analysed studies (n = 10) that met the eligibility criteria of their research in the pursuit of determining the impact school-based PA initiatives had on children’s (n = 2617, aged 6.04–15.04 years) enjoyment levels when participating in PA. The results suggest that participation in the PA initiatives evaluated had a small but significant (*p* = 0.01) effect on children’s PA enjoyment [17]. Remmers et al. [18] suggested that PA enjoyment may be an important determinant in children achieving and maintaining positive habitual PA patterns over time, highlighting the influential impact school-based PA initiatives may have on children’s PA behaviour. A systematic literature review (SLR) of school-based PA interventions (n = 31) [19] reported that participation in school-based PA initiatives reduces symptoms of anxiety and significantly improves feelings of resilience (*p* = 0.001), wellbeing (*p* = 0.001) and positive mental health (*p* = 0.001) in children (aged 8–17 years old). The impact that long-term (12–72 months) participation in school-based interventions, which included a PA component, had on primary school children’s (n = 22,381) BMI was examined in an SLR conducted by Mei et al. [20]. Intervention participants exhibited significantly (2.23 kg/m less; *p* < 0.05) lower increases in BMI scores when compared to control participants [20]. Furthermore, interventions that involved <100 min of weekly PA significantly (*p* < 0.05) reduced the level of BMI growth in participants when compared to interventions comprising of >100 min per week [20], suggesting that short sessions of well-designed PA initiatives are capable of positively affecting children’s body composition. These results illustrate the potential benefit carefully designed school-based initiatives may have on children’s health-related metrics. However, considerable heterogeneity was illustrated between the studies analysed in the research [17,19,20], suggesting that the findings should be interpreted cautiously.

TDM was initially developed by the headteacher of St. Ninians’s primary school in Scotland to combat the children’s perceived lack of fitness [21,22]. The dissemination of TDM from the setting of its original inception has resulted in the adoption of this PA initiative in over 16,000 schools across 90 countries [23]. It has been reported that 22.1% of state-funded primary schools in England have registered to participate in TDM [24]. Moreover, it was reported that larger schools located in urban areas with higher percentages of disadvantaged pupils were more likely to adopt and implement TDM [24]. According to the principles published by The Daily Mile Foundation [25], TDM is a simple, inclusive and non-competitive initiative that requires no equipment to implement and offers child participants the opportunity to engage in PA for 15 min during every school day. When participating in TDM, children are encouraged to run, or walk when needed to catch their breath, around a route that has a mud-free surface [25]. In addition, the implementation of TDM has been designed to prevent any additional workload on teachers’ schedules. However, and most important of all, it has been reported to positively impact the physical and mental health and wellbeing of participating children [23].

Emerging evidence suggests that TDM has been successfully implemented and sustained in various primary school settings [21,26,27,28,29,30]. Furthermore, participation in TDM appears to positively affect markers of children’s health [26,30,31,32,33,34]. However, Thorburn [22] argues that these perceived benefits are largely based on anecdotal evidence and advocates for the implementation of wide-ranging research studies to ascertain whether such benefits are supported by robust methodological and data collection procedures. A recent SLR has shown the effect of TDM on metrics such as PA, physical health, mental health, wellbeing, academic performance and cognitive function with results suggesting that participation in TDM has the potential to positively impact both a child’s PA behaviour and cardiorespiratory fitness (CRF) [35]. The same authors highlight how the implementation of quality and robust methodologies are required in future research to further the understanding of the complex relationships that exist between TDM and participating children’s health-related metrics (e.g., cognitive function). However, despite the extensive research conducted by Breslin et al. [35], common facilitators and barriers associated with effective TDM implementation were not explored and thus, warrant further investigation.

This current review serves to critically analyse the research that has been published on TDM to date, with the expressed objectives to: (1) identify the impact that participating in TDM has on a range of primary school children’s health-related metrics; (2) evaluate the implementation process of TDM in selected journal articles to explore the impact contrasting school, teacher and class group characteristics have on delivery; (3) identify the presence and impact of limitations associated with TDM research studies; and (4) present practical implications and recommendations to enhance the sustainability capacity of TDM in primary school settings.

## 2. Materials and Methods

### 2.1. Search Strategy

The study identification and screening process for inclusion, which adheres to the Preferred Reporting Items for Systematic Reviews and Meta-Analyses (PRISMA) [36], is illustrated in Figure 1. A three-phase approach was implemented to systematically search for all articles relating to TDM initiative [37]. The first phase involved searching EBSCO, PubMed and ScienceDirect databases for all articles related to TDM initiative from the years 2012–2022. The preliminary search was conducted using the phrase ‘The Daily Mile’, while the second phase involved the expansion of the search on the EBSCO, PubMed and ScienceDirect databases to include terms closely related to common themes identified across the articles retrieved from the preliminary search, such as ‘cardiorespiratory fitness’, ‘body composition’, ‘physical activity (PA)’, ‘cognition’, and ‘process evaluation’. The search continued until the saturation point was reached, whereby no new studies were identified. The initial two phases, which involved searching the databases, resulted in the identification of 202 journal articles selected for further analysis.

The third phase involved the references of every article retrieved from the first two phases of the systematic search being uploaded to Mendeley Reference Manager [38] for storage, before being exported to Rayyan, a web-based literature screening tool [39]. Duplicate articles were subsequently removed (n = 91). The remaining articles (n = 111) were assessed on their suitability for further inclusion. The screening of titles and abstracts facilitated the identification and removal of 83 articles that did not meet the eligibility criteria of this review. These articles were excluded because they did not analyse the implementation, long-term uptake or effectiveness of TDM (n = 76), an abstract or title to the article was not displayed (n = 4), the research design of the article was a Systematic Literature Review (SLR) (n = 1), or further duplicates were identified during the screening process (n = 2). Following the conclusion of the screening process, the remaining articles (n = 28) were reviewed in full to determine their eligibility to meet the inclusion criteria. It was ultimately determined that 16 articles from the initial database search met the eligibility criteria of this review. Following the completion of the third phase of the systematic search strategy and the final selection of studies that met the eligibility criteria (n = 16), reference lists of the appropriate articles were reviewed to explore additional relevant studies that may not have been considered in the initial review process. However, no additional studies were determined to be eligible for inclusion in this review.

### 2.2. Eligibility Criteria

Studies involving an evaluation of the implementation, long-term uptake and/or effectiveness of TDM initiative were selected for inclusion. Longitudinal, cross-sectional, observational, experimental or quasi-experimental randomised controlled trials (RCTs) that evaluated TDM within a primary school setting were eligible for inclusion. Research that adopted a mixed-method approach, combining study designs presented above, were also deemed eligible for inclusion. Finally, studies were excluded if (i) they were not published in English, (ii) they were published before the year 2012, (iii) the research was conducted outside a primary school setting, (iv) they were not related to the implementation, long-term uptake or effectiveness of TDM initiative, and/or (v) were not published in a peer-review journal.

### 2.3. Study Characteristics

An analysis of the methods and outcomes presented within the included studies (n = 16) was conducted, which resulted in the identification of five distinct outcome categories in TDM research: (1) cardiorespiratory fitness (CRF), (2) anthropometry and body composition, (3) PA behaviour, (4) cognition and (5) process evaluation of TDM. The methodologies of the eligible studies are reviewed before their findings are grouped and discussed under the five outcome categories.

### 2.4. Research Context

Eleven of the 16 studies measured the impact of TDM on health-related metrics of primary school children. Table 1 presents a breakdown of the number of selected studies that measure one of the five distinct outcomes. Nine studies (56%) examined the implementation process of TDM or measured a metric related to its long-term uptake in primary school settings. The impact of TDM on CRF was the most researched health-related metric (5 out of 16 studies—31%), followed by anthropometry and body composition (4 out of 16 studies—25%), PA behaviour (4 out of 16 studies—25%) and cognition (3 out of 16 studies—19%). The majority of studies were conducted in settings within the United Kingdom (UK). Holland and Italy are the only two countries outside of a UK context to evaluate the implementation or effectiveness of TDM (Table 2). 

### 2.5. Research Design and Data Collection Methods

Table 2 presents a breakdown of the research design and data collection methods implemented in each study included in this review (n = 16). Thirteen studies deemed eligible for inclusion involved the participation of primary school children ranging in age from 6–11 years old. Primary school staff members, including principals, teachers and classroom assistants, participated in 8 of the studies included for review. Both experimental and observational studies were included. Eight studies (50%) implemented an experimental design within their research. Every experimental study included a control group for comparison, with the exception of the study conducted by Marchant et al. [30]. There were 10 (62.5%) observational designs identified from the 16 studies. Three research studies adopted a mixed-method approach comprising both an experimental and observational design [26,29,40].

**Table 2 ijerph-20-06203-t002:** Breakdown of research designs and data collection methods.

Authors	Study Design	Participants	Location	Quantitative Methods	Qualitative Methods	Did Participating Schools Have Daily Mile (TDM) Experience?	Frequency of TDM Delivery Measured?
Booth et al. (2020) [33].	Cross-sectional repeated measures	Children (n = 5463; boys = 2702 and girls 2761; mean age 10.2 years).	England, Scotland, Wales and Northern Ireland.	Stop-signal task, static boxes search task, reading span task, Adapted Children’s Feeling Scale and Felt Arousal Scale *.	N/A	No	N/A
Breheny et al. (2020) [41].	Cluster randomised controlled trial (cRCT)	Children ** experimental (exp.) arm (n = 1153; mean age, 8.8 ± 1.1; 47.6% were girls) and a control arm (n = 1127; mean age, 8.8 ± 1; 47.5% were girls).	Birmingham, England.	Body Mass Index z-score (BMIz), body fat %, linear track test ◦, quality-adjusted life years (QALYs) ◦◦. Child Health Utility 9 Dimension (CHU9D), Middle Years Development Instrument (MDI) ◦◦◦.	N/A	No	No
Brustio et al. (2020) [33].	Quasi-exp.	Children: An exp. arm implemented TDM 3 times a week (n = 111; mean age, 9.19 years; 46.8% were girls), an exp. arm implemented TDM 2 times a week (n = 168; mean age, 8.91 years; 50.6% were girls) and a control arm (n = 269; mean age, 9.26 years; 49.1% were girls)	Turin, Italy.	BMI, Waist to Height Ratio (WHtR) and 6 Minute Run Test (6MRT).	N/A	No	Yes
Brustio et al. (2019) [26].	Quasi-exp.	Phase one: children: exp. arm (n = 486; mean age, 8 ± 1 years; 45.7% were girls) and a control arm (n = 309; (mean age, 8 ± 1 years; 49.8% were girls). Phase two: teachers (n = 33; mean age, 45 ± 5 years; 100% were women).	Turin, Italy.	Phase one: BMI, WHtR, 6 min. run test and standing long jump □. Phase two: survey questionnaire.	N/A	No	Yes
Chesham et al. (2018) [31].	Quasi-exp.	Children: exp. arm (n = 252, 123 were girls; mean age, 8.1 ± 2.0 years) and a control arm (n = 127, 64 were girls; mean age, 8.8 ± 1.8 years).	Stirling, Scotland.	Accelerometer examined moderate-vigorous physical activity (MVPA) and sedentary time □□, 20 Metre Shuttle Run Test (20M SRT), skinfolds and BMIz.	N/A	No	No
de Jonge et al. (2020) [34].	Quasi-exp.	Children: combined exp. arms (n = 253, 50.2% were girls; mean age, 10.0 ± 0.1 years) and control arm (n = 338, 52.1% were girls; mean age, 10.1 ± 0.1 years).	Holland.	20M SRT.	N/A	No	Yes
Hanckel et al. (2019) [27].	Rapid ethnographic assessment	Public health practitioners, headteachers, assistant headteachers, teachers and children (n = 63). Overall, 56% of participants were female and 44% male.	London, England.	N/A	Interviews (n = 22), focus groups (n = 11) and participant observations (61 h) of TDM implementation process.	Yes, all participating schools (n = 5) had experience of implementing TDM.	Yes
Harris et al. (2020) [28].	Multi-method process evaluation	Phase one: children (n = 75, 42.7% were girls; mean age, 7 years 8 months). Phase two: key stage one children (n = 5, 100% were boys; mean age, 6 years 11 months), key stage two children (n = 6, 67% were girls; mean age, 9 years 1 month), teachers (n = 4, 100% were female), parents (n = 2, 100% were female) and parent-governor (n = 1, female).	East Midlands, England.	Phase one: self-report logs, OMNI perceived exertion scale and system for observing fitness instruction time (SOFIT).	Phase two: focus groups (n = 4).	No	Yes
Hatch et al. (2021) [42].	Descriptive cross-sectional	Year 5 children (n = 52, 29 were girls; mean age, 10.1 ± 0.2 years) and year 6 children (n = 20, 9 were girls; mean age, 11.3 ± 0.3 years).	East Midlands, England.	20M SRT ##, heart rate ^ and global positioning systems (GPS) assessed PA.	N/A	Six participating schools had experience of implementing TDM. Two participating schools had no experience of implementing TDM.	N/A
Hatch et al. (2021) [40].	Within-subject counterbalanced randomised crossover trial	Children (n = 104); boys (n = 56; mean age, 10.4 ± 0.7 years) and girls (n = 48; mean age, 10.4 ± 0.7 years).	East Midlands, England.	Stroop test, Sternberg paradigm and Flanker test.	Focus groups (n = 87 children).	Six participating schools had experience of implementing TDM. Two participating schools had no experience of implementing TDM.	N/A
Malden and Doi (2019) [29].	Interpretive process evaluation	Teachers (n = 13, 11 were women).	Edinburgh and East Lothian, Scotland.	N/A	Semi-structured interviews.	Yes, all participating teachers (n = 13) had experience of implementing TDM in their school.	No
Marchant et al. (2020) [30].	Mixed-methods natural exp.	Phase one: Headteachers, teachers and children ^^ Phase two: Baseline (n = 225, 117 were boys; mean age, 10.2 ± 1.0 years) and follow-up (n = 232, 130 were boys; mean age, 10.6 ± 0.6 years) #.	Wales	Phase two: 20M SRT	Phase one: interviews (n = 11) and focus groups (n = 6).	No	No
Morris et al. (2019) [43].	RCT	Children: exp. arm (n = 158, 88 were girls; mean age, 8.99 ± 0.5 years) and a control arm (n = 145, 82 were girls; mean age, 8.99 ± 0.5 years).	Yorkshire, England.	Accelerometer assessed PA, Trail Making Test (TMT), Digit Recall, Flanker task, Animal Stroop task and Maths Addition and Subtraction, Speed and Accuracy Test (MASSAT).	N/A	Yes, all participating children (n = 303) were from schools (n = 11) experienced with the implementation of TDM.	N/A
Routen et al. (2021) [44].	Mixed-method process evaluation	Phase one: Primary school TDM implementors (n = 25) and primary school non-implementors of TDM (n = 17). Phase two: teaching staff (n = 5).	Leicester, England.	Phase one: survey questionnaire.	Phase two: interviews (n = 5).	There was a mix of schools from Phase one that implemented (n = 25) and did not implement TDM (n = 17). Three schools from phase two implemented TDM, while 2 did not or had ceased implementation.	Yes
Ryde et al. (2018) [21].	Process evaluation	Current headteachers (n = 3), teachers (n = 1), classroom assistant (n = 1) and the retired headteacher who created TDM (n = 1).	Central Scotland.	N/A	Semi-structured interviews (n = 6).	Yes, all participants (n = 6) were from schools that had experience of implementing TDM.	Yes
Ward and Scott (2019) [45].	Process evaluation	Teachers (n = 4) and children (n = 12).	West Midlands, England.	N/A	Interviews (n = 4), focus groups (n = 4) and observations.	Yes, the participating school (n = 1) had experience of implementing TDM.	No

* The mediating effect wellbeing had on the relationship between PA intensity and cognition was measured. ** 2280 children provided data at baseline before three schools dropped out. ◦ CRF results from this study were not included for analysis in this review as missing data exceeded 56% for the participating sample. ◦◦ An economic analysis of TDM was conducted using QALYs which was beyond the scope of this review. ◦◦◦ High volumes of participant data missing for pre and post-test measures of CHU9D and MDI ensures that these results cannot be used to predict an accurate interpretation of TDM’s effect. □ The muscle power of participants’ lower extremities was assessed by standing long jump performance which was beyond the scope of this review. □□ Assessing the effect TDM has on sedentary behaviour was beyond the scope of this review. ^ Assessing the effect TDM has on heart rate was beyond the scope of this review. ^^ Headteachers, year 5 and 6 teachers and pupils were invited to participate but the exact number of final participants is unclear. # The % increase in 20M SRT performance was calculated from the change between pre- (68% of sample participated) and post- assessment (70% of sample participated). ## 20M SRT performance was measured to calculate and determine the impact CRF has on PA behaviour during TDM participation.

## 3. Results

The results of the five outcome categories selected to assess the impact of The Daily Mile (TDM) research are below. These categories are classified as (1) cardiorespiratory fitness, (2) anthropometry and body composition, (3) physical activity behaviour, (4) cognition and (5) TDM process evaluation. The interdependent nature of each of these five categories is recognised despite each category being examined individually within this section. The presentation of results from each study included in this review precedes a discussion related to identified limitations and recommendations for future research. These sections are related to each of the previously mentioned five outcome categories.

### 3.1. Cardiorespiratory Fitness (CRF)

Five papers (31%) examined the impact of TDM on CRF (Table 3) and each of the five studies attributed the physical activity initiative (PA) initiative to have a positive impact on measures of CRF [26,30,31,33,34]. Furthermore, 4/5 of these studies included intervention and control groups, and relative to the control group, the intervention group displayed greater improvements [26,31,33,34].

The frequency of delivery was monitored in 3/5 studies [26,33,34]. Brustio et al. [33] evaluated the dose-effect relationship between TDM and children’s CRF over a period of 6 months and found that the group that participated in TDM three times a week demonstrated a higher percentage increase in distance covered from pre- to post- in a 6-minute Run Test (6MRT) (+8.8%) in comparison to the control group (+6.2%) and the group participating in TDM twice a week (+5.6%). Findings collected by de Jonge et al. [34] suggested that exposure to TDM for a period of 12 weeks promotes CRF improvements, as TDM participants exhibited a significantly greater increase (*p* < 0.01) in 20 Metre Shuttle Run Test (20M SRT) scores from pre- to post- assessments when compared to a control group (+0.8 vs. +0.1). Moreover, results suggest that additional support, including personal school visits from health promotion experts and regular reminder messages to implement TDM, does not facilitate increased CRF improvements, as the change in 20M SRT scores demonstrated within TDM intervention arm were significantly greater than the changes exhibited in 20M SRT scores within TDM intervention arm that was allocated additional personal support from JOGG (+1.1 vs. +0.6; *p* < 0.01), co-ordinators and promoters of TDM in Holland. The intervention and intervention with the additional personal support group were expected to perform TDM on any day that their physical education (PE) classes were not scheduled [34]. The implementation rate of TDM in the intervention classes was reported to be 88%, while classes from the intervention-plus group were documented to have delivered TDM 90% of the prescribed time [34]. Brustio et al. [26] found that TDM participants significantly increased the distance they covered in the 6MRT from pre- to post- test (+25.15 m, *p* < 0.001; percent change = 3.1%) in comparison to a control group (+4.44 m, *p* = 0.911; percent change = 0.5%). Chesham et al. [31] concluded that participation in TDM was associated with a significant improvement in the distance covered from a pre- to post-20M SRT (+39 m; *p* = 0.037) in comparison to a control arm (Table 3). Moreover, Marchant et al. [30] reported that children from high and low socio-economic groups participating in TDM exhibited similar increases in 20M SRT scores from pre- to post- assessment (low socio-economic: 23.8 [±17.1]—28.4 [±18.3]; high socio-economic: 34.8 [±19.6]—39.8 [±20.9]) following participation in TDM. However, the implementation rate was not directly assessed in two studies [30,31], which may have hidden TDM’s true value effect on CRF. 

### 3.2. Anthropometry and Body Composition

Four studies (25%) evaluated the impact of TDM on children’s anthropometric and body composition measures (see Table 4). Of the included studies, each one measured the effect on body mass index (BMI), two (12.5%) measured the effect on waist-to-height ratio (WHtR) and two (12.5%) analysed the impact of TDM participation on measures of body fat percentage [26,31,33,41]. Furthermore, 2/4 studies monitored the dose–effect relationship between TDM and measured outcomes [26,33].

The intervention and control arms displayed similar insignificant levels of change to their BMI in each of the four studies [26,31,33,41]. Furthermore, the difference in the level of change of WHtR measures between intervention and control arms was not significant in the studies of Brustio [26] and Brustio et al. [33]. Chesham et al. [31] demonstrated how TDM arm exhibited significantly greater reductions in measures of body fat percentages (−1.4 mm; *p* = 0.034). In contrast, the intervention (21.9–22.2%) and control groups (21.8–22.3%) in the study conducted by Breheny et al. [41] displayed similar marginal increases in body percentages from pre- to post- assessments. However, after adjusting for covariates such as school size, pre-test school mean BMIz and the percentage of children eligible for free school meals, it was concluded that the difference in the level of change between groups at the 12-month follow-up was significant (*p* = 0.049). 

### 3.3. PA Behaviour

Four studies (25%) evaluated the relationship between TDM and children’s PA behaviour (Table 5). Two of these four studies used accelerometers to measure children’s PA output [31,43], one used GPS [42], while one measured PA behaviour via direct observation methods [28].

Chesham et al. [31] evaluated the impact of TDM on children’s habitual PA and concluded that participation in the initiative resulted in significant improvements to children’s MVPA (+9.1 min) in comparison with a control group. According to Harris et al. [28], children engaged in a high percentage (+88.1–100%) of MVPA during the completion of a TDM session. In addition, Morris et al. [43] reported that children accumulated a significantly higher amount of MVPA (+10.67 ± 2.74 min) during a TDM session relative to a control group who continued with academic lessons (0.44 ± 0.95 min).

Hatch et al. [42] determined that children covered a mean distance of 2511 ± 606 m, covered the greatest distance at low and moderate speeds (i.e., speed zones 2 and 3) and engaged in intermittent PA behaviour (i.e., mean number of entries into five different speed zones was 646 ± 175) when participating in TDM. Moreover, boys covered a significantly greater mean distance compared to girls (2717 ± 606 m vs. 2305 ± 398 m; *p* = 0.001), girls covered a significantly greater distance in low-speed running (speed zone 2: 0.84–2.84 m/s) compared to boys (1469 ± 365 m vs. 1272 ± 396 m), boys covered a significantly greater distance in moderate-speed (speed zone 3: 2.85–3.79 m/s; boys: 1095 ± 784 m, girls: 560 ± 361 m) and high-speed running (speed zone 4: 3.80–4.73 m/s; boys: 233 ± 161 m, girls: 156 ± 126 m) and boys engaged in significantly more intermittent PA during TDM relative to girls (*p* < 0.001) [42]. Furthermore, children in fitness quartile 4 (highest fitness) covered a significantly greater distance than children from quartile 1 (mean difference = 828 m; *p* < 0.001), quartile 2 (mean difference = 673 m; *p* < 0.001) and quartile 3 (mean difference = 494 m; *p* = 0.015), while quartile 4 covered a significantly greater distance in moderate-speed running relative to all other fitness quartiles (quartile 1 vs. 4: *p* < 0.001; quartile 2 vs. 4: *p* = 0.001; quartile 3 vs. 4: *p* < 0.001) and the fittest (quartile 4) and least fit (quartile 1) engaged in the most intermittent PA; however, the difference between all fitness groups was not significant (*p* = 0.198) [42]. 

### 3.4. Cognition

Executive function processes were measured by all three studies (19%) that aimed to evaluate the relationship between TDM and children’s cognition (Table 6). Contrasting tools and methods were used by each study, whereby children completed paper-based tests such as the Trail Making Task (TMT), Digit Recall, Flanker test (FT) and the Animal Stroop task in the study of Morris et al. [43]. Booth et al. [32] had children complete computer-based tests such as a Stop-Signal task, Static Boxes Search task and a Reading Span task and Hatch et al. [40] had children complete a Stroop test, Sternberg Paradigm (SP) and FT on a computer.

Booth et al. [32] determined that TDM had a significant impact on inhibition (*p* = 0.00) and verbal working memory (VWM) (*p* = 0.00), but not visual-spatial working memory (VSWM) (*p* = 0.165) when compared to a control group that rested outdoors. Hatch et al. [40] did not attribute TDM as having a significant effect on the response time or accuracy for any level of the Stroop test, response time for the one-item and three-item levels or accuracy for the one-item, three-item and five item levels of the SP, or the response time for the congruent and incongruent levels of the FT. However, accuracy was reported to be significantly higher for the TDM trial on both the congruent (97.3 ± 0.3% vs. 96.0 ± 0.5%; (*p* = 0.011) and incongruent (93.9 ± 0.5% vs. 92.2 ± 0.8%; *p* = 0.023) levels of the FT, and responses were significantly slower for the TDM trial (972 ± 19 ms) compared to the control trial (937 ± 20 ms) on the five-item SP (*p* = 0.030) [40]. Moreover, although not significant (*p* = 0.099), accuracy tended to be higher on the SP five-item level for TDM trial (87.1% vs. 85.7%), which suggests that TDM may have a direct impact on the reaction time and indirect impact on the accuracy associated with children’s working memory. In contrast, Morris et al. [43] found TDM did not have a significant impact on the total scores (i.e., correct answers minus errors) for any of the executive function tests including the TMT-A (*p* = 0.356), TMT-B (*p* = 0.813), standard FT (*p* = 0.448), Mixed FT (*p* = 0.527) or the Animal Stroop task (*p* = 0.894) when compared to the control group. However, the value displayed for the interaction of statistical significance between the Digit Recall (*p* = 0.019) test and trial conditions (i.e., TDM and control) contradicts the report that “no significant interactions were found in any of the executive functions total scores” [43] (p. 5). 

### 3.5. Process Evaluation

Nine studies (56%) included in this review measured an element related to the TDM implementation process. A variety of tools were used by each study to evaluate the implementation process of TDM, whereby semi-structured interviews (n = 6) were the most frequently used tool [21,27,29,30,44,45], followed by focus groups (n = 5) [27,28,30,40,45] observations (n = 2) [27,45] and surveys (n = 2) [26,44] and analysis of secondary data (n = 1) [27]. A breakdown of the data collection processes and presentation of key findings from research studies that analysed TDM implementation process can be found in Table 7. 

#### 3.5.1. Facilitators

TDM is perceived to attribute improvements to children’s physical fitness, physical literacy and PA behaviour, which encourages the implementation of the initiative by relevant stakeholders and substantiates its delivery [21,27,28,29,30,40,44]. In addition, delivery of TDM promotes and fosters healthy social interactions between children participating and facilitates the development of a healthy social rapport between pupils and teachers [27,28,29,30,40]. Moreover, participating in TDM can positively impact children’s behavioural characteristics such as concentration and attention levels when they return to class [27,28,40,44]. In addition, TDM was perceived by some teachers to positively impact children’s concentration levels in two further studies [29,30]. According to Marchant et al. [30], TDM’s ability to positively affect participant’s concentration and behaviour during class-time is largely based on the characteristics of the child. However, participating in TDM is perceived to positively affect components of children’s psychological health such as wellbeing, self-esteem, stress relief and academic competency [30]. In support, the perceived need for, and realisation of, various benefits has been reported to contribute towards the adoption, implementation, and long-term uptake of school-based PA initiatives [48].

The importance of teachers’ self-enrolment and belief in the benefits of participating in TDM motivates children to partake and engage with the initiative [28,30,44]. Furthermore, participation from teachers in TDM sessions appears to be strongly associated with increased engagement levels for child participants [28,29,30,44]. As mentioned by a Key Stage 2 (KS2) student participating in a focus group in the research of Harris (p. 10) [28],

“When the teachers did it, I started to run more than I did at first.”

According to Ryde et al. [21], teachers’ participation in TDM occurred at the school where it originated and was viewed by most participants as a complementary component associated with the implementation of the initiative.

A school that values the health and wellbeing of its children, combined with a culture encompassing an approach tailored towards providing PA and sporting opportunities, facilitates the continued implementation of TDM [44]. Moreover, the autonomy and flexibility afforded to teachers regarding the integration of the delivery of TDM into teaching schedules contributes towards the successful implementation of the initiative [21,28]. According to Erwin et al. [49], allocating teachers with control and leadership over the delivery facilitates the subtle integration of an intervention into a teacher’s schedule. In support, a teacher participating in the study conducted by Ward and Scott [45] rebranded the name of TDM in their school to ‘******* (name of school) Run’ to support teachers’ autonomy and reduce pressures associated with having to implement TDM every day. Comparably, some schools participating in the study by Hanckel et al. [27] used different terms such as ‘The Daily Run’ or ‘The Daily Stroll and Chat’ when referring to TDM. Furthermore, the ability to adapt TDM to suit their schools’ context and characteristics of their children was seen by teachers as a favourable component associated with the implementation of the initiative [21,28,30,44]. Moreover, tailoring the implementation of TDM to the needs and traits of the school and children, and prioritising the delivery of PA opportunities for children over how TDM is implemented, will inevitably result in the original format of the initiative being adapted [44]. These adaptations include designing pathways on school playgrounds, delivery of different types of PA that can be implemented during inclement weather notwithstanding the dearth of a sheltered outdoor area, introducing competitive elements, and ‘keeping TDM fresh’ by introducing new components to how TDM is implemented at the start of each school year and different variations to deliver TDM on a daily basis (e.g., use of tokens) [44]. An example of how TDM’s original format can be adapted and developed was discussed by one interviewee participating in the research of Routen et al. [44], (p. 6),

“we’ve created our own version to keep the children engaged. You see with the skipping, there’s so many different variations. They can learn tricks; they can do spins. Like we’ve got a few big ropes so they’re doing different games, different activities with the big ropes, they just love having little competitions against each other to see who can skip for the longest, it’s just a little bit more, they can vary it whereas with TDM you can’t really.”

According to the principles of The Daily Mile Foundation [25], children should be aware that TDM is not a competition. However, findings from the research suggest that teachers often support and encourage self-competition and competition between peers to motivate and increase children’s engagement levels in TDM [27,28,29,44]. According to Harris et al. [28], some parents expressed concern regarding the maintenance of TDM in their child’s school if motivating techniques were not implemented and children were deterred from competing with one another. Incentivising children to participate in TDM, through various methods such as using tokens to the track the number of laps completed and rewarding progress, was attributed as being as an important facilitator in the continued implementation of the initiative [21,27,29,30,44]. Furthermore, linking the implementation of TDM to curriculum topics and/or external events enhances learning opportunities and ensures the initiative stays relevant to the needs and desires of children [21,27].

#### 3.5.2. Barriers

Busy teaching schedules and the competing demands associated with existing curriculum expectations present a considerable barrier to the implementation of TDM [27,28,29,30,44]. In support, time constraints have been reported as the most significant barrier associated with the implementation of PA interventions in primary and secondary schools [50]. In addition, PE acts as barrier towards the implementation of TDM, as the initiative is often not delivered on days when PE is scheduled to take place [21,27,29]. The teacher’s schedule, in addition to other competing school-based activities, largely impacts whether the TDM is implemented on a given day or not [27].

Inappropriate clothing and footwear during wet and icy conditions were identified as factors that can hinder the delivery of TDM [21,29,30,44]. The weather was seen as a barrier for participation by participants in the study conducted by Hanckel et al. [27] only if it resulted in unsafe implementation conditions (e.g., heavy rain, icy conditions, etc.). However, children’s weather preferences have been shown to influence their level of enjoyment when participating in TDM [40]. Moreover, the time taken to get children ready to partake in TDM often exceeds 15 min, resulting in additional time constraints associated with implementing the initiative [27,28,29,44]. In low socio-economic areas, children regularly attend school with clothing and footwear not suited to allow safe participation in TDM [21,44]. Subsequently, the implementation of TDM is often negatively impacted in low-socioeconomic areas during wet weather conditions as many children have unsuitable or inappropriate resources to participate (e.g., no jacket, shoes with holes, etc.) [21]. Furthermore, hot weather was highlighted in one particular study by a teacher with regard to potentially posing additional health and safety concerns (e.g., dehydration, exhaustion, etc.) for children when participating in TDM [30].

Feedback collected from relevant stakeholders in the study conducted by Harris et al. [28] suggests the physical environment and facilities of the school present barriers to teachers’ engagement and parents’ support for TDM. The authors reported that teachers encouraged children to change into wellington boots when TDM is implemented on a grass field during the winter months, which subsequently presents additional time constraints. Furthermore, qualitative data collated by Malden and Doi [29] indicate that grass areas are not suitable to implement TDM during wet conditions due to associated safety concerns (i.e., slip hazards). A lack of and inadequate spaces were identified as primary implementation barriers by schools that do not implement TDM [44]. Moreover, space constraints associated with the school environment necessitate the adaptation of TDM for implementation [27]. However, the implementation of TDM at an external facility creates additional logistical issues, as a risk assessment of the route has to be carried out each day and extra staff would be required to supervise the children [21]. In addition, the repetitive nature of TDM can leave children feeling bored and unmotivated to participate [40]. In support, the absence of a variety of modes of PA and strict adherence to the original TDM format has been attributed to lower enthusiasm levels and disengagement among children when participating in the initiative [30].

## 4. Discussion

### 4.1. Limitations of TDM Research

Schools were not randomly assigned to the control condition in four studies that compared the impact TDM has on a specific metric with a control arm [26,31,33,34]. Varied group characteristics can cause selection bias if participants are not randomly assigned to their research condition [51]. Without the random assignment of participants to each condition, it cannot be concluded that identified and unidentified biases are disseminated consistently between experimental and control arms [52]. Furthermore, the absence of a control arm in the study conducted by Marchant et al. [30] magnified the difficulties associated with determining TDM’s impact on children’s CRF. The control arm acts as a comparison for what would have happened if the experimental arm was not exposed to the intervention, reducing the risk of bias and isolating the intervention’s impact in the process [53].

The dose of delivery was not monitored in three studies that examined the effect of TDM on various health-related metrics from pre- to post- tests [30,31,41]. The attribution of an outcome effect to an intervention requires careful and precise monitoring of the dose received by participants [54]. In addition, there were considerable amounts of missing or invalid data collected for some measured outcomes between pre- to post- tests in some studies [26,31,41]. Essentially, the statistical power of a research study is reduced by missing data and often results in biased evaluations and invalid interpretations of the effectiveness of a measured outcome [55].

Unequal dose-effect conditions were generated by Chesham et al. [31] as a result of the varied duration of time that elapsed between the pre- and post-assessments for the intervention and control arms (i.e., 7 vs. 3 months). In addition, data was collected during different seasonal timeframes for the comparison arms [31]. According to Kukull and Ganguli [56], information bias exists when the outcome or exposure is measured differently between participating groups (e.g., length of time) and the statistical significance of the findings are diminished when such bias is present. Similarly, schools participating in TDM experimental condition in the research of Marchant et al. [30], were exposed to TDM for varied lengths of time during the pre- and post-CRF assessments. Furthermore, data were collected in two different phases during different seasons, as participating schools decided when to begin implementing TDM [30]. The season [57] and subsequent weather conditions [58] affect PA behaviour and may have influenced the results produced by Chesham et al. [31] and Marchant et al. [30] as a result.

In four controlled TDM experimental studies, the control arm was significantly older than the participating intervention arms at baseline [26,31,33,34]. This variance in age between groups may have potentially biased CRF results, as findings presented in the research of Roberts et al. [59] suggest that performance in the 20M SRT by children aged 9–10 and 11–12 years old is affected by relative age effects across each age group. Furthermore, children’s weight and height increase between the ages of 5–18 years old, and this subsequently results in considerable rises in their BMI calculation [60]. As a result, the imbalance in age illustrated across experimental and control arms in three studies may have skewed the true value outcome effect that TDM has on children’s anthropometric and body composition measures [26,31,32]. Furthermore, the control arms performed significantly better in baseline CRF tests in three of the studies that measured the relationship between TDM and CRF [26,31,34]. According to Roberts and Torgerson [61], reported outcome differences between groups may be a result of varying group characteristics, not exposure to an intervention. As a result, any outcome attributed to TDM, which is based on findings produced by studies that demonstrate a baseline imbalance in relevant characteristics, must be interpreted with caution.

The application of objective PA measures over subjective methods in child studies are recommended due to the intermittent nature of their PA patterns [62]. The effect that TDM has on children’s MVPA patterns presented in the research by Harris et al. [28] must therefore be interpreted cautiously due to the observational method implemented. According to McKenzie [47], approximately 12% of all ‘System for Observing Fitness Instruction Time’ (SOFIT) assessments should be coded concurrently by two independent observers to ensure reliability in methodology. Subsequently, the findings reported by Harris et al. [28] may not have been reliable due to the observation being conducted by a solitary observer. In addition, the requirement for children to wear an accelerometer for eight consecutive days in the study by Chesham et al. [31] may have led to misleading conclusions, as factors guiding PA behaviour on weekends cannot be accredited to TDM [63]. Furthermore, accelerometer data was collected in 60 s epochs by Chesham et al. [31], which, according to the research of Aibar and Chanal [64], may demonstrate higher bouts of light and moderate PA but underestimate the time children spend engaged in vigorous PA.

The teachers of class groups participating in the study by Booth et al. [32] decided the order and timing of the delivery of the outdoor activities. As a result, order effects and teachers’ instructions to children while participating in the outdoor activities may have unintentionally led to biased results [32]. In the research conducted by Morris et al. [43], there were classes that implemented TDM precisely according to the time-focused 15-min principle (m = 15 ± 0.0 min) while other classes were instructed to implement TDM as they usually would (m = 14.07 ± 3.01 min). Subsequently, the varied duration of participation in TDM may have resulted in unintentional information bias being present [56]. In addition, TDM experimental trial implemented in the research of two studies [40,42] continued for a duration of 20 min, which differs from one of the TDM’s core guidelines that the initiative should take no longer than 15 min to complete [25]. Subsequently, any effect, or lack of, may not represent the impact of a TDM session with a duration of 15 min.

### 4.2. Practical Implications and Recommendations

The comparison of outcome results across studies that analyse the impact of a solitary PA initiative such as TDM is often problematic as a result of the implementation of varying methods of measurement [65]. As stated by Cavill et al. [66], the evaluation of various public health interventions is often impeded by a lack of consistency in the modes of measurement. In support, Dickersin and Mayo-Wilson [67] recommend the implementation and adoption of standardised data collection processes to facilitate the ease of comparing results across clinical research studies. Moreover, adhering to shared design standards is essential in identifying the existence of a correlation between two variables [67]. It is recommended, therefore, that future researchers consistently adopt reliable and valid measurement tools used in previous TDM-related studies, to facilitate the comparison of results across research studies that evaluate TDM’s impact on various metrics.

Future researchers should strive to implement a randomised controlled trial (RCT) design when analysing the effectiveness of TDM initiative. RCTs are regarded as the gold standard in study design for evaluating interventions due to their ability to minimise or prevent bias [68]. The random assignment of participants to various study conditions reduces the risk of selection bias resulting from variances in group properties [68]. Moreover, the effectiveness of TDM can be determined through the implementation of well-designed RCTs and comparison of outcome changes between intervention and control groups from pre- to post-assessments [68,69]. However, the logistical challenges associated with conducting research within a school setting, including acquiring permission and support from school staff [70,71], ensure researchers are likely to encounter considerable difficulties when endeavouring to implement an RCT design within a primary school setting.

Research analysing TDM’s relationship with health-related variables must strive to include a measure of the dose that is delivered to participants, to ensure outcome effects can be assuredly attributed to TDM [54]. A measure of the implementation fidelity of initiatives such as TDM is required in evaluation studies so that any identified effect, or lack hereof, can be accurately attributed to participation in the initiative [72,73]. Moreover, Hayes et al. [74] conclude that the successful implementation of school-based obesity preventative interventions is facilitated by empowering teachers with the opportunity to adapt components of the interventions. To facilitate long-term engagement and maintenance of the initiative, the adaptation and development of TDM’s original format should be encouraged within schools and class groups that encounter common implementation barriers (i.e., inadequate facilities). However, caution must be exercised when adapting elements of an intervention to ensure its ‘spirit’ or effectiveness are not compromised in the process [75]. Subsequently, it is essential that future researchers strive to determine the components of TDM initiative that are fundamental to the initiative’s success and long-term uptake in primary schools through programme differentiation examination [73].

It is recommended that future studies evaluating the impact of TDM on physical outcomes are conducted over the course of at least a full academic year (i.e., September–June) to facilitate the identification of the true value TDM participation has on health-related metrics. Children’s CRF remains relatively constant over time and is unlikely to deviate too much month-to-month [76,77]. In support, results collected from the research of Alves et al. [78], which analysed the impact that 8 weeks of strength and aerobic training had on children’s (n = 168; aged 10–11 years old, 10.9 ± 0.5) explosive strength and maximal oxygen uptake (VO2max), suggest long periods of sustained exercise and training are likely to induce significant increases in children’s CRF capacity. The experimental condition that incorporated strength and aerobic training within the one session (1.7 ± 1.9%; *p* = 0.00) and the experimental condition that completed strength and aerobic trainings on separate days (3.1 ± 1.5%; *p* = 0.00) experienced significant VO2max improvements from pre-to post- test when compared to a control arm (0.2 ± 1.6%; *p* = 0.386) [78]. Furthermore, the research of Jackson and Cunningham [79] reported that the BMI of American elementary school children (n = 4938) remained relatively stable from kindergarten to 8th Grade (0.79–0.85), which suggests TDM is unlikely to significantly impact children’s BMI after a short period of time (i.e., 3 months).

Finally, the findings from a recently published single-arm pilot study by Arkesteyn et al. [80] suggest participation in TDM has the potential to positively impact children’s wellbeing. Future research should strive to implement quality research study designs to further explore the relationship between TDM and markers of children’s wellbeing. In addition, there is an opportunity for future researchers to investigate and evaluate the effect of combining elements of a classroom-based curriculum with the delivery of TDM to determine the influence it has on long-term uptake of the initiative. In support, Cassar et al. [48] promote the design of interventions that alter the teaching style (e.g., physical activity combined with cognitive tasks) as opposed to interventions that require changes to the curriculum to be successfully implemented. The interpretation of results from this future study has the potential to alter how TDM is delivered, improving children’s health and wellbeing and promoting the long-term uptake of the initiative in the process.

## 5. Conclusions

Results presented from the included articles suggest participating in TDM positively impacts markers of a variety of health-related metrics including CRF and PA behaviour. Future research should strive to bridge the gap in evidence that explores the relationship between TDM and children’s wellbeing and quality of life. The maintenance of TDM in primary school settings may require bespoke amendments to the original format to overcome implementation barriers such as time constraints and inadequate facilities. Moreover, optimising children’s enjoyment levels, stimulating their interest and maintaining their long-term engagement with the initiative may also require additional modifications to the original format of TDM to nullify the barrier effect its repetitive nature has on implementation.

## Figures and Tables

**Figure 1 ijerph-20-06203-f001:**
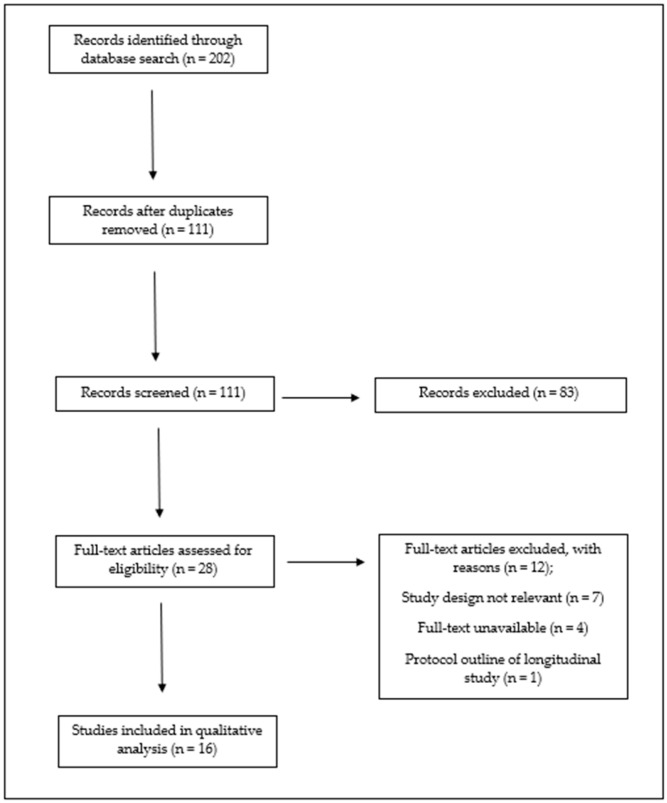
Search strategy and identification of appropriate studies.

**Table 1 ijerph-20-06203-t001:** Breakdown of studies included by outcome measured.

Outcome Measure	Number of Studies
Cardiorespiratory Fitness (CRF)	5
Anthropometry and Body Composition	4
PA Behaviour	4
Cognition	3
Process Evaluation	9

**Table 3 ijerph-20-06203-t003:** TDM research methodology and results related to CRF.

Authors	Study Length	Study Design	Participants	Method of Measurement	Dose Response Reported	TDM Effect on CRF
Chesham et al. (2018) [31].	7 months *	Quasi-exp.	A TDM exp. school (n = 240) and a control school (n = 117).	20M SRT	No	Significant increase in exp. arm’s level of change when compared to the control arm (*p* = 0.037) ×.
Brustio et al. (2019) [26].	3 months	Quasi-exp.	A TDM exp. arm (n = 486) and a control arm (n = 309) #	6MRT	Yes	Significant difference (*p* = 0.026) exhibited in the level of change between the exp. arm (+3.1%) and the control arm (+0.5%) ^
Brustio et al. (2020) [33].	6 months	Quasi-exp.	An exp. arm implemented TDM 3 times a week (n = 111), an exp. arm implemented TDM 2 times a week (n = 168) and a control arm (n = 269).	6MRT	Yes	3 times TDM exp. arm exhibited significant improvements (+8.8%). 2 times TDM exp. arm exhibited significant improvements (+5.6%). Control arm exhibited significant improvements (+6.2%).
de Jonge et al. (2020) [34].	3 months	Quasi-exp.	A TDM exp. arm (n = 82), a TDM exp. + Support (n = 146) arm and a control arm (n = 308)	20M SRT <	Yes	TDM Exp. arm exhibited significant improvements (+17.4%). TDM Exp. + support arm exhibited significant improvements (+10.6%). The control arm showed no significant improvement. Participants in the exp. arms exhibited significantly greater improvements (+11.9%) when compared to the control arm. □
Marchant et al. (2020) [30].	6 months	Uncontrolled-exp.	68% of the sample participated in the baseline CRF assessment (n = 229) and 70% participated in the follow-up assessment (n = 235)	20M SRT	No	TDM exp. arm improved mean 20M SRT scores from baseline to follow-up tests (30.7 ± 19.3—35.5 ± 20.5). No significant difference demonstrated in the level of change between children from high and low socio-economic groups (High: 34.8 ± 19.6—39.8 ± 20.9; Low: 23.8 ± 17.1—28.4 ± 18.3). >

* Varied exposure lengths between groups in study. × adjusted for age, sex and age * sex. # 70.7% and 47.2% of the students in the experimental and control groups completed the 6MRT. ^ Adjusted for age and sex. < SRT conducted over 18 m due to limitations associated with data collection locations. □ Adjusted for age and sex. > Adjusted for age and sex, and the % increase in SRT performance was calculated from the change between pre- (68% of sample participated) and post- assessment (70% of sample participated).

**Table 4 ijerph-20-06203-t004:** TDM research methodology and results related to anthropometry and body composition.

Authors	Study Length	Study Design	Participants	Method of Measurement	Dose Response Reported	Effect
Chesham et al. (2018) [31].	* 7 months	Quasi-exp.	A TDM exp. school (n = 246) and a control school (n = 123).	BMIz	No	No significant difference exhibited in the level of change (*p* = 0.785) between both conditions. ^
Chesham et al. (2018) [31].	* 7 months	Quasi-exp.	A TDM exp. school (n = 213) and a control school n = 114).	Sum of triceps, biceps, iliac crest and subscapular skinfolds	No	The exp. arm exhibited a significant (*p* = 0.034) reduction (−1.4 mm) when compared to the control arm. ^
Brustio et al. (2019) [26].	3 months	Quasi-exp.	A TDM exp. arm (n = 486) and a control arm (n = 309) ~	BMI	Yes	Both conditions exhibited the same insignificant level of change (−0.6%). <
Brustio et al. (2019) [26].	3 months	Quasi-exp.	A TDM exp. arm (n = 486) and a control arm (n = 309) ~	WHtR	Yes	No significant difference in the level of change (*p* = 0.529) between the exp. (+2.2%) and control (0.0%) arms. <
Brustio et al. (2020) [33].	6 months	Quasi-exp.	An exp. arm implemented TDM 3 times a week (n = 111), an exp. arm implemented TDM 2 times a week (n = 168) and a control arm (n = 269).	BMI	Yes	No significant difference (*p* = 0.275) in the level of change between the exp. arm implementing TDM 3 times per week (+0.7%), the exp. arm implementing TDM 2 times per week (+1.7%) and the control arm (+4.2%).
Brustio et al. (2020) [33].	6 months	Quasi-exp.	An exp. arm implemented TDM 3 times a week (n = 111), an exp. arm implemented TDM 2 times a week (n = 168) and a control arm (n = 269).	WHtR	Yes	No significant difference in the level of change between the exp. arm implementing TDM 3 times per week (−1.2%), the exp. arm implementing TDM 2 times per week (+1.7%) and the control group (+4.2%).
Breheny et al. (2020) [41].	12 months	Cluster RCT	A TDM exp. arm (n = 850) and a control arm (n = 820).	BMIz	No	No significant difference (*p* = 0.146). in the level of change between TDM exp. arm (+17.77%) and the control arm (+19.15%). A moderate and significant effect (*p* = 0.001) × was found in girl participants, but not in boys.
Breheny et al. (2020) [41].	12 months	Cluster RCT	A TDM exp. arm (n = 850) and a control arm (n = 820).	A measure of participants’ body fat % was calculated using a Tanita bioimpedance monitor.	No	The control arm (+2.24%) exhibited a significant (*p* = 0.049) × increase in the level of change when compared to TDM exp. arm (+1.35%). Significant differences were exhibited between girls and boys (*p* = 0.001).

* Varied exposure lengths between groups in the study, as the intervention group was exposed to TDM conditions for roughly 7 months and approximately 3 months elapsed between the pre and post-test measurements for the control group. ^ Adjusted for age and sex and age * sex. < Adjusted for age and sex × Adjusted for school size, % free school meals, school BMIz, school baseline outcome, participant baseline outcome. ~ 91.4% of the exp. group and 94.9% of the control group provided data for pre- and post- measurements.

**Table 5 ijerph-20-06203-t005:** TDM research methodology and results related to PA behaviour.

Authors	Study Length	Study Design	Participants	Method of Measurement	Dose Response Reported	Effect
Chesham et al. (2018) [31].	* 7 months	Quasi-exp.	A TDM exp. school (n = 56) and a control school (n = 62).	Accelerometers ~, Evenson cut points [46] and a 60 s epoch length.	No	TDM exp. arm exhibited significant improvements in MVPA minutes accumulated (+9.1 min) compared to the control arm. #
Harris et al. (2020) [28].	3 months	Quasi-exp.	A TDM exp. school (Key Stage 1 [KS1] students n = 42; Key Stage 2 [KS2] students n = 33).	SOFIT observation [47] conducted in Week 12 of the study by one observer.	Yes	KS1 participants spent 100% of a TDM session engaged in MVPA. KS2 participants spent 88.1% of a TDM session engaged in MVPA.
Morris et al. (2019) [43].	N/A ◦	Cross-sectional RCT	A TDM exp. arm (n = 158) and a control arm (n = 145).	Accelerometers, Evenson cut points [46] and a 15 s epoch length.	N/A ◦	TDM exp. arm engaged in significantly greater amounts of MVPA (10.67 ± 2.74 min) when compared to the control arm (0.44 ± 0.95 min). (*p* = 0.001).
Hatch et al. (2021) [42].	N/A ^	Descriptive cross-sectional	A TDM exp. arm (n = 72)	GPS	N/A ^	The mean distance covered during TDM was 2511 ± 550 m. The highest distance was covered during the first 5 min (748 ± 141 m). Boys covered a significantly greater distance than girls (2717 ± 606 m vs. 2305 ± 398 m; *p* = 0.001). Distance covered walking (*p* = 0.001), at moderate speed (*p* < 0.001) and high speed (*p* < 0.001) varied significantly over time. PA patterns were most intermittent during the first 5 min, with boys engaging in significantly more intermittent PA patterns throughout the 20 min than girls (*p* < 0.001).

* Varied exposure lengths between groups in the study, as the intervention group was exposed to TDM conditions for approximately 7 months and approximately 3 months elapsed between the pre- and post-test measurements for the control group. ◦ A cross-sectional design was implemented to assess participants’ PA behaviour during a TDM session. ~ Data collected from accelerometers worn by participants for eight consecutive days was used to measure PA behaviour. A valid measurement required at least 10 h of wear time for 3 days. # Following adjustment for age, sex and age. ^ A cross-sectional design was implemented to evaluate PA patterns during a TDM session.

**Table 6 ijerph-20-06203-t006:** TDM research methodology and results related to cognition.

Authors	Study Length	Study Design	Participants	Method of Measurement	Dose Response Reported	Cognition Effect
Booth et al. (2020) [32].	N/A ◦	A Repeated measure design (RMD) to compare and evaluate the acute impact of participating in TDM, PA of maximal effort (20M SRT) and a classroom break with no exercise (Control).	Children (n = 5463) from 332 primary schools participated.	An adapted stop-signal task measured inhibition. An adapted static boxes search task measured visual-spatial working memory (VSWM). A reading span task measured verbal working memory (VWM) An adapted version of the children’s feeling scale and felt arousal scale measured the mediating impact of wellbeing.	N/A ◦	TDM had a sig. effect on inhibition (*p* = 0.000) and VWM (*p* = 0.000) but not on VSWM (*p* = 0.165) when compared to the control condition. TDM had a sig. effect on inhibition (*p* = 0.000), VWM (*p* = 0.000) and VSWM (*p* = 0.010) when compared to the 20M SRT. Affect (i.e., element of wellbeing test) had a sig. mediating effect on children’s VWM (*p* = 0.0) and an insignificant mediating effect on inhibition (*p* = 0.595) and VSWM (*p* = 0.324). Alertness (*p* = 0.008) (i.e., element of Wellbeing test) had a sig. mediating effect on children’s VWM (*p* = 0.008) and an insignificant mediating effect on inhibition (*p* = 0.727) and VSWM (*p* = 0.796).
Morris et al. (2019) [43].	N/A #	Cross-sectional RCT	A TDM exp. arm (n = 158) and a control group (n = 145).	The Trail Making Task (TMT), Digit Recall (DR), Flanker Task (FT) and Animal Stroop Task measured inhibitory control and working memory. The Maths Addition and Subtraction, Speed and Accuracy Test (MASSAT) measured maths fluency.	N/A #	No sig. effects found between TDM and control arms for the total scores in TMT-A (*p* = 0.356), TMT-B (*p* = 0.813), standard FT (*p* = 0.448), Mixed FT (*p* = 0.527), the Animal Stroop task (*p* = 0.894) or DR (*p* = 0.019) × tests. TDM arm exhibited insignificant MASSAT improvements (*p* = 0.136) from pre- to post- test when compared to the control arm.
Hatch et al. (2021) [40].	7 days ^	Within-subject counterbalanced randomised crossover trial	Children (n = 104) from 8 primary schools participated.	The Stroop test (ST), Sternberg paradigm (SP) and Flanker task (FT) measured inhibitory control, visual working memory and cognitive flexibility components.	N/A ^	No sig. difference found between TDM and control trials for response times on the simple (*p* = 0.605) or complex (*p* = 0.520) levels or accuracy on the simple (*p* = 0.873) or complex (*p* = 0.885) levels of the ST. However, accuracy in the complex level of the ST was inclined to be higher immediately following TDM participation when compared to the control trial (*p* = 0.057). No sig. difference found between TDM and control trials for response times on the one-item level (*p* = 0.661) or three-item level (*p* = 0.143) of the SP. Responses were sig. slower in the TDM trial (972 ± 19 ms) compared to the control trial (937 ± 20 ms) on the five-item level of the SP (*p* = 0.030). Accuracy was similar across both trials on the one-item level (*p* = 0.235) and three-item level (*p* = 0.700) of the SP. However, accuracy was inclined to be higher immediately following TDM trial relative to the control trial (*p* = 0.073) on the one-item level. Accuracy was also inclined to be higher during the TDM trial (87.1 ± 0.9%) relative to the control trial (85.7% ± 1.1%) on the five-item level of the SP (*p* = 0.099). No sig. difference found between TDM and control trials for response times on the congruent level (*p* = 0.980) or incongruent level (*p* = 0.537) of the FT. Accuracy was sig. higher for the TDM trial on both the congruent level (97.3 ± 0.3% vs. 96.0 ± 0.5%; (*p* = 0.011) and incongruent level (93.9 ± 0.5% vs. 92.2 ± 0.8%; *p* = 0.023) of the FT. The effect of TDM on response times or accuracy tasks was not sig. influenced by gender or fitness across any cognition test, with the exception of the SP three-item level whereby gender sig. influenced (*p* = 0.027) the effect of TDM on response time.

◦ Booth et al. [32] compared the acute effect participation in TDM has on cognition and wellbeing with different classroom breaks. # Morris et al. [43] implemented a cross-sectional design to identify the acute effect TDM has on executive function and academic performance. × The reported *p*-value appears to contradict the report that no executive function test was significantly affected by either TDM or control trial. ^ Seven days separated the completion of each trial (i.e., TDM and control trials).

**Table 7 ijerph-20-06203-t007:** TDM research methodology and results related to TDM process evaluation.

Authors	Geographical Location of School(s)	Schools (N =)	Participants	Method of Measurement	Key Findings
Ryde et al. (2018) [21].	Central Scotland	4	Head teacher, teachers, and classroom assistant (n = 6)	Semi-structured Interviews	Simple central principles, flexible delivery and the adaptability of components facilitate the successful implementation of TDM.
Brustio et al. (2019) [26].	Turin, Italy	5	Teachers (n = 33)	Survey	TDM was implemented on average of 3 times per week. Most teachers (81–96%) perceived TDM to have a positive impact on components associated with the school’s environment. Most teachers (62–93%) perceived TDM as easy to implement. 72% of teachers were happy to participate in TDM. 50% of teachers indicated that they would like to continue with TDM in the following school year.
Harris et al. (2020) [28].	East Midlands, UK	1	Children, parents, teachers and principal (n = 18)	Focus groups	Flexibility and autonomy regarding the delivery of TDM was important to teachers. The school’s facilities, time taken by children to change into appropriate footwear and clothing and the seasonal timing of delivery (i.e., during winter) were identified as critical implementation barriers. Teachers’ encouragement and participation in TDM and the inclusion of competition were identified as implementation facilitators.
Hanckel et al. (2019) [27].	Lewisham, South London, England	5	Public health practitioners, headteachers, assistant headteachers, teachers and children (n = 63)	Interviews, focus groups, observations and secondary data analysis	TDM was implemented 2–4 times per week. The inclusion of games and competition supports engagement levels. Time demands associated with curriculum pressures and school-based activities were identified as critical implementation barriers. The implementation of TDM was regularly adapted and varied between schools and classroom-to-classroom.
Malden and Doi (2019) [29].	Edinburgh and East Lothian, Scotland	8	Teachers (n = 13)	Semi-structured interviews	Most teachers used incentives to increase children’s motivation and engagement levels with TDM. Inadequate facilities and inappropriate clothing were identified as implementation barriers. The time-consuming nature and subsequent effect on learning time acted as a barrier to implementation.
Ward and Scott (2019) [45].	West Midlands, UK	1	Teachers and children (n = 16)	Observations, interviews and focus groups	The teachers’ schedule influenced when TDM was delivered. TDM was implemented mainly as a classroom break and in response to the teachers’ duty of care to the children’s physical health. Prioritising other activities in the school facilitated the long-term uptake of TDM. Teachers dictated when and where TDM was implemented while children and teachers both influenced how it was implemented.
Marchant et al. (2020) [30].	South Wales	6	Headteachers, teachers and children *	Semi-structured interviews and focus groups	The flexibility to adapt the original format of TDM creates a positive experience. Teachers’ promotion and active participation in TDM contributes to children’s motivation levels. The weather and conditions associated were identified as implementation barriers. Competition motivates some children but also leads to lower engagement levels with others.
Hatch et al. (2021) [40].	East Midlands, UK	8	Children (n = 87)	Focus groups	Core principles of TDM such as exercising outdoors, engaging with peers and the self-paced nature that supports autonomy positively impact children’s enjoyment of the initiative. TDM is perceived to have a positive impact on concentration and learning and health-metrics such as children’s fitness. The perceived/actual exercise ability of children and their weather preferences influence their enjoyment of TDM. The majority of children expressed a desire for a greater variety of exercise elements when participating in TDM as running every day becomes boring and repetitive.
Routen et al. (2021) [44].	Leicester, England	Phase 1 (n = 42) and phase 2 (n = 5)	Phase 1: TDM coordinators (n = 25) and Physical Education leads (i.e., TDM non-implementors) (n = 17). Phase 2: TDM coordinators (n = 5)	Phase 1: Survey Phase 2: Interviews	59.5% of schools that participated in the survey implement TDM, of which the majority (96.6%) delivered TDM ≥3 days each week. Timetable and curriculum demands, space and safety concerns, namely caused by adverse weather conditions emerged as critical implementation barriers. Teachers’ involvement and buy-in to TDM positively impacts children’s engagement with the initiative. Prioritising the opportunity for children to engage with PA daily, whereby TDM is adapted to suit the characteristics of the school, class groups and children, over how TDM is implemented was identified as an important measure of success.

* Headteachers, year 5 and 6 teachers and pupils were invited to participate but the exact number of final participants is unclear.

## Data Availability

No new data were created or evaluated in this study. Sharing data is not applicable to this article.

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
