# Peer review of "A Systematic Review of the Implementation and Effectiveness of ‘The Daily Mile’ on Markers of Children’s Health"

_ijerph, 2023, doi:10.3390/ijerph20136203_

Round 1
Reviewer 1 Report
Thank you for the opportunity to review – A Systematic review of the implementation and Effectiveness of ‘The Daily Mile’ on markers of children’s health.
I compliment the authors on a well organised and clearly written paper. The research is relevant and provides important context to further understand the implementation of focused PA with children. Presented is a good range of relevant literature with clear examples across a wide context. Overall, a very professionally written introduction and justification for the systematic review.
The review itself was conducted thoroughly, using a PRISMA methodology. I do think that the results represent outcomes from the 16 studies rather than true thematic analysis. Therefore to describe these as outcome themes may need further consideration as your overall results reflect ‘content’ and may be better presented as outcome categories.
Further points for consideration
1. Table 2 – Consider moving survey type response studies from the Qualitative Methods column into the Quantitative column. See entries for Brustio et al. Phase 2 (Quasi – exp) and Routen et al. Phase one.
2. Include a good depiction of The Daily Mile in the Introduction. There are many descriptors about it, however no detail about what the children actually do. This would give context to statements about changing the ‘repetitive’ nature. Readers who are not familiar with the TDM should be able to see what the children actually undertake?
3. Table 2 – Column 7. Did Participants have Daily Mile Experience? Would consider rewording this to make it clear and specific to which participants – staff or students? It is unclear as written.
4. Suggest adding the year of publication with the authors for the studies presented in the Tables.
Author Response
"Please see the attachment"

Reviewer 2 Report
Thanks for having me review the paper, and few comments are as follows:
1. could you please describe the reasons why you need to exclude the paper from original records screend specifically (Figure 1)
2. Table 1: it might be good to show the how many times of positive or negative effects on those outcome measures in addition to show numbers of studies.
Author Response
"Please see the attachment"

Reviewer 3 Report
The introduction provides an appropriate context for the need for physical activity to improve children's health and well-being and the dangers of physical inactivity, highlighting WHO recommendations.
The objectives are clearly defined following a review of the scientific literature.
The PRISMA method is one of the internationally preferred tools for systematic reviews and meta-analyses and is therefore a good methodological choice.
On the other hand, the contribution of studies from outside the UK provides greater consistency to the review.
The results and discussion of the analyses performed are well presented and provide sufficient information.
The tables provide sufficient and well ordered and structured information that allows for an optimal assessment of the different methods used and the effects achieved in each of the studies.
Practical implications and recommendations
The conclusions are in line with the objectives proposed in this study.
The bibliographical references are relevant, sufficient and up to date, based on the nature of the work carried out.
Author Response
"Please see the attachment"
